# Effect of CO_2_ on the Desulfurization of Sintering Flue Gas with Hydrated Lime

**DOI:** 10.3390/ma16010303

**Published:** 2022-12-28

**Authors:** Jianguo Hong, Xinqing Zou, Ziqiang Qin, Bin Zhou, Shuhua Geng, Yuwen Zhang, Xingli Zou, Xionggang Lu

**Affiliations:** 1State Key Laboratory of Advanced Special Steel & Shanghai Key Laboratory of Advanced Ferrometallurgy & School of Materials Science and Engineering, Shanghai University, Shanghai 200444, China; 2Ironmaking Plant, Shanghai Meishan Iron and Steel Co., Ltd., Nanjing 210039, China

**Keywords:** sintering flue gas, semi-dry desulfurization, desulfurization, calcium carbonate, ultra-low emissions

## Abstract

The effect of carbon dioxide (CO_2_) on the desulfurization of sintering flue gas with hydrate (Ca(OH)_2_) as an absorbent was investigated, and the formation of calcium carbonate (CaCO_3_) and its effect on the desulfurization was discussed. The competitive relationship between carbon dioxide (CO_2_) and sulfur dioxide (SO_2_) with the deacidification agent in sintering flue gas is discussed thermodynamically, showing that sulfates are more likely to be generated under high oxygen potential conditions and that SO_2_ reacts more preferentially than CO_2_ under a thermodynamic standard state. The amount of produced CaCO_3_ increases under the condition that the CO_2_ concentration is absolutely dominant to SO_2_ in the sintering flue gas atmosphere. The effect of temperature, humidity and CO_2_ concentration on the desulfurization of Ca(OH)_2_ are discussed experimentally. The increasing temperature is not conducive to desulfurization, and the humidity can promote desulfurization, while excessive humidity could inhibit desulfurization. The suitable relative humidity is 20%. In situ generated calcium carbonate has a certain desulfurization effect, but the desulfurization effect is not as good as Ca(OH)_2_. However, a large proportion of CaCO_3_ was produced in the desulfurization ash under the condition that CO_2_ concentration was absolutely dominant to SO_2_ in the sintering flue gas atmosphere.

## 1. Introduction

The sintering process is a major source of sulfur and nitrogen pollutant emissions of steel mills, and the proportion of SO_2_ and NOx account for about 60% and 50% of the total emissions from the iron and steel industry, respectively [1,2]. In recent years, the Chinese government strengthened environmental protections, and a series of stricter environmental laws and regulations have been published. With the proposed new concept of “Ultra-low emission”, the SO_2_ and NOx emission standards were changed to <35 mg/Nm^3^ and <50 mg/Nm^3^, respectively [3], which were ≤200 mg/Nm^3^ and ≤300 mg/Nm^3^ in the 2015 national standard [2].

To match the emission requirements, many methods were applied in iron and steel factories, and the main methods include wet desulfurization technology [4,5,6,7,8], dry desulfurization technology [9,10] and semi-dry desulfurization technology [9]. The semi-dry desulfurization technology has drawn a lot of attention for its advantages of multi-pollutant synergistic control, water saving and low investment in equipment [11]. The mainstream denitrification technologies are SCR (selected catalytic reduction) [12], SNCR (selected non-catalytic reduction) [13], activated carbon adsorption [14] and ozone oxidation–adsorption [11]. Compared with SCR, SNCR and activated carbon methods, the ozone oxidation–absorption method exhibits the advantages of high oxidation selectivity, high oxidation efficiency, and no secondary by-products [15].

Shanghai Meishan Iron and Steel Co. adopted the semi-dry desulfurization technology for simultaneous desulfurization and denitrification from flue gas using a flue gas circulating fluidized bed. Ozone (O_3_) was introduced to oxidize NO into dioxide (NO_2_), which is soluble nitrogen and highly active [16]. In the absorption tower, the main component of hydrated lime is Ca(OH)_2_, the desulfurization rate of sintering flue gas reaches 98% and the denitrification rate reaches 88%, and those meet the requirements of ultra-low emissions for sintering flue gas pollutants.

However, there is a significant increase in the desulfurization ash production under the “ultra-low emission” working conditions, from the original 1349.8/t to 2101.6/t, an increase of about 55.7%. Meanwhile, the composition of CaCO_3_ in desulfurization ash also increased from 31.17% to 51.50%, an increase of 65.2%, which not only increased the usage of hydrated lime but also imposed a burden on waste disposal and risked potential environmental pollution. Therefore, it is necessary to know the mechanism of CaCO_3_ increase under a high oxide atmosphere.

Hydrated lime is an adsorbent commonly used in semi-dry desulfurization; NO_2_ and SO_2_ react with hydrated lime, curing to form nitrates and sulfates [17,18,19,20,21,22,23,24]. The main components of the sintering flue gas are SO_2_ 1026~1321 mg/Nm^3^, NOx 271.4~293.5 mg/Nm^3^, O_2_ 15.5~16.5 vol%, CO_2_ 6.1~7.0, H_2_O 10.1~10.9 vol% and dust 28.6~39.0 mg/Nm^3^. The components of CO_2_ contribute to the generation of a large amount of CaCO_3_. For a long time, CO_2_ has not received attention, and there are few studies on the change in CaCO_3_ in the desulfurization ash. CaCO_3_ as an adsorbent is mainly used in the high-temperature stage of 800~1200 °C, which decomposes to CaO to complete the desulfurization reaction [25,26]. For the sintering flue gas temperature conditions (80~120 °C), whether CaCO_3_ has the effect of desulfurization in the absorption tower has not been clarified.

In this paper, hydrate was selected as the absorbent because it is widely used in the semi-dry desulphurization process in steel works. CO_2_ contained in the sintering flue gas is a very important factor that needs to be studied. A study focusing on the effect of CO_2_ in desulfurization was conducted. The competitive reaction of the gas components SO_2_ and CO_2_ in sintering flue gas was analyzed thermodynamically. Under the simulated sintering flue gas condition, the effect of CO_2_ concentration, temperature and relative humidity on the desulfurization of calcium-based adsorbent were carried out. The mechanism of CaCO_3_ formation and its effect of desulfurization under low temperatures were investigated. The research results will help to understand the mechanism of the formation of desulfurization ash under ultra-low emission conditions.

## 2. Materials and Methods

### 2.1. Experimental Materials

The raw material used in this paper is quicklime, provided by Meishan Iron and Steel Plant (Nanjing, China). The powdered raw material obtained after grinding and crushing was added to deionized water for digestion, and then the finished suspension was dried to constant weight, and finally the adsorbent was crushed and sieved. As shown in Figure 1a, the original quicklime was mainly composed of CaO, CaCO_3_ and Ca(OH)_2_, of which the main ingredient was CaO. The main component of the digested sample was Ca(OH)_2_ (Figure 1b). The results of laser particle size analysis (Figure 1c) showed that the particle size distribution of original quicklime was 0.5~100 µm, and the average particle size was 7.8 µm, which was consistent with the SEM analysis results (Figure 1d), and the particle sizes were not uniform. The flow chart of making the samples is shown in Figure 1e. A certain amount of samples was taken, and it was ground into powder for XRF element analysis test. The test results are shown in Table 1 It can be seen from the table that raw quicklime contains elements such as Ca, Mg, Fe, Al, Si and K (Table 1). As shown in Table 1, the total calcium content is 88.26 wt.%. The TG-DSC curve of the original quicklime sample is shown in Figure 2, which was obtained in an Ar atmosphere. It is clear that the mass loss process occurs in two different stages. The first stage is at 422.2 °C, which is attributed to the decomposition of Ca(OH)_2_, while the second stage, at 647.3 °C, shows the decomposition of CaCO_3_. According to the mass loss rates Δw_1_ and Δw_2_ of the two temperature segments on the thermogravimetric map, the mass fractions of Ca(OH)_2_ and CaCO_3_ can be roughly obtained; the calculation formula of Ca(OH)_2_ content is w = Δw_1_ × 74/18, and the formula of CaCO_3_ content is w = Δw_2_×100/44. Therefore, we found that the Ca(OH)_2_ content is 75.44 wt.% while the CaCO_3_ content is 12.43 wt.%. The specific surface area of raw quicklime was 3.3850 m^2^/g (Figure 3a), with a relatively uniform pore size distribution and an isothermal adsorption curve close to type IV with an obvious H3-type hysteresis loop.

### 2.2. Experimental Methods

#### 2.2.1. Experimental Device

The schematic diagram of the experimental system is shown in Figure 4. The high-purity gases (N_2_, SO_2_ and CO_2_) from the high-pressure cylinder are mixed in the mixing tank. The specific purities of gases are listed in Table 2. The composition of the simulated blast furnace gas is controlled by mass flow controller (MFC). The simulated flue gas reacted with the calcium-based adsorbent in the tubular heating furnace. The gas composition was detected by a Testo 350 flue gas analyzer. The gas from the measuring system then passed through the tail gas absorption with the Na(OH)_2_ solution to remove the acidic components of the tail gas.

#### 2.2.2. Experimental Methods

1.Relative humidity calculation

Previous studies have shown that Ca(OH)_2_ cannot react directly with SO_2_ in an absolutely dry environment [27,28], so water is introduced into the desulfurization reaction by the water vapor carrying method.

The principle of this method is based on Dalton’s law of partial pressure, and the relative humidity is controlled by adjusting the temperature of the constant temperature water bath. The water bath is heated to a set temperature, maintained for a certain time, and the inert gas N_2_ was introduced into the water through the pipeline and bubbled in the water to saturate the water vapor, so as to obtain the humidity conditions required for the experiment. The specific calculation of relative humidity is shown in Equation (1).
(1)$$H=\frac{0.622p\delta }{p_{0}-p\delta }$$

H—Absolute humidity in grams per kilogram (g∙kg^−1^).

P—Saturation vapor pressure of water in Pa (Pa).

Δ—Relative humidity in percent (%).

P0—The total pressure of wet air in Pa (Pa).

2.Desulfurization ash composition analysis

Since there is no corresponding standard for the analysis of desulfurization ash components, the component analysis methods are mainly established based on GB176-1996 “Methods of Analysis of Cement” and GB/T5484 “Methods of Chemical Analysis of Gypsum”. The calcium hydroxide content and calcium carbonate content are quantitatively analyzed by thermogravimetry, and the calcium sulfite content is determined by the iodometric method.

#### 2.2.3. Evaluations

The evaluations of the desulfurization effect of adsorbent in the experiments are defined as follows.

1.Desulfurization efficiency

The specific calculation of desulfurization efficiency is shown in Equation (2).
(2)$$\eta_{So_{2}}=\frac{C_{SO_{2},in}-C_{SO_{2},out}}{C_{SO_{2},in}}\times 100\%$$

η—The desulfurization efficiency (%).

$C_{SO_{2},in}$—The concentration of SO_2_ in the inlet reactor flue gas, mg/Nm^3^.

$C_{SO_{2},out}$—The concentration of SO_2_ in the outlet reactor flue gas, mg/Nm^3^.

2.Penetration time

When the SO_2_ concentration of tail gas is higher than 35 mg/Nm^3^, which is the limitation of the national standard, the adsorbent is penetrated and ineffective, and the elapsed time is defined as the adsorbent penetration time.

3.Penetration sulfur capacity

The ratio of the absorbed sulfur mass by penetration to the original adsorbent mass is defined as the theoretical sulfur capacity. The specific calculation of theoretical sulfur capacity is shown in Equation (3).
(3)$$\mathrm{Ps}=\frac{\mathrm{Ms}}{\mathrm{Mt}}\times 100$$

P_S_—The penetration sulfur capacity, g/100 g adsorbent.

Ms—The absorbed sulfur mass by penetration, g.

Mt—The original adsorbent mass, g.

#### 2.2.4. Characterization Methods

The particle size distribution (PSD) tests of raw materials were carried out by a laser particle size analyzer (Malvern PANalytical, Mastersizer 2000 MU, The Netherland). The phase composition was characterized by X-ray diffraction (XRD, Bruker-AXS D8 Advance, Billerica, MA, USA) using Cu Kα radiation (λ = 1.54056 Å) at a scan rate of 4° min^−1^ from 10 to 80° (2θ). The microscopic observations and analyses were performed by scanning electron microscope (SEM, FEI Nova Nano SEM 450, Hillsboro, OR, USA). Before being transferred into the test chamber, the specimens were sputtered with gold coating on the surface. Thermal analysis was conducted using thermogravimetric analysis combined with differential scanning calorimetry (TG-DSC, NETZSCH STA449 F3, Selb, Germany). The purge gas used for TG-DSC was N_2_, and the heating rate was set to 10 °C/min. The temperature rise range was set at 100~1100 °C.

Gas compositions were analyzed by a Testo 350 flue gas analyzer. The device is widely used in the analysis of plant flue gas for its small size, high sensitivity and portability. Meanwhile, it can measure a variety of complex gas components at the same time without interference. The analyzer adopted electrochemical principles, using the gas at the corresponding oxidation potential for potential electrolysis, and then calculated the concentration of the gas by the current consumed.

## 3. Results and Discussion

### 3.1. Thermodynamic Calculation

The main components of the sintering flue gas are N_2_, CO_2_ and CO, and the main composition of the adsorbent is Ca(OH)_2_. In the temperature range of the sintering flue gas (80~120 °C), N_2_ and CO do not react with hydrated lime and are treated as inert gases. The desulfurization reaction of the sintering flue gas is mainly the reaction of SO_2_ with Ca(OH)_2_, as shown in reactions (4) and (5). In the flue gas, there is about 6~7 vol% CO_2_, which can also react with Ca(OH)_2_ (reaction (6)), forming a competitive relationship with SO_2_. Figure 5 shows the ΔG^θ^-T relationship of these chemical reactions. In addition, the generated CaCO_3_ has a removal effect on SO_2_, which occurs in reactions (7) and (8). Thermodynamically, the desulfurization capacity of Ca(OH)_2_ is higher than that of CaCO_3_. In the presence of high oxygen potential, it is easier to produce CaSO_4_, a stable sulfate, which is more favorable for SO_2_ removal.
*Ca(OH)_2_(s)* + *SO_2_(g)* = *CaSO_3_* + *H_2_O*(4)
*2Ca(OH)_2_* + *2SO_2_(g)* + *O_2_(g)* = *2CaSO_4_* + *2H_2_O(g)*(5)
*2Ca(OH)_2_* + *2CO_2_(g)* = *2CaCO_3_* + *2H_2_O(g)*(6)
*CaCO_3_* + *SO_2_(g)* = *CaSO_3_* + *CO_2_(g)*(7)
*CaCO_3_* + *SO_2_(g)* + *1/2O_2_(g)* = *CaSO_4_* + *CO_2_(g)*(8)

### 3.2. Effect of Temperature on Desulfurization Performance of Adsorbent

In the absorption tower, the temperatures of the sintering flue gas at the entrance and exit are about 120 and 80 °C, respectively. To simulate the actual temperature of the absorption tower, the desulfurization temperatures were set as 80 °C, 100 °C and 120 °C, respectively. The carrier gas N_2_ flow rate was set to 1000 mL/min, the SO_2_ flow rate was set to 500 mL/min, and the adsorbent loading was 2 g. Figure 6 shows the desulfurization curves of the adsorbent at different temperatures.

It can be seen from Figure 6 that the increase in desulfurization temperature was unfavorable for the desulfurization process, and the penetration time became shorter with the increase in temperature. The penetration time was 573 s under 80 °C and 471 s at 120 °C, with a difference of 102 s. As the temperature increased, the desulfurization capacity was decreased, and the utilization rate of the adsorbent decreased. This is due to the fact that the increase in temperature reduces the solubility of SO_2_ in water, which in turn reduces the concentration of SO_2_ on the surface of the adsorbent, making the sulfur capacity lower.

### 3.3. Effect of CO_2_ Concentration on Desulfurization Performance of Adsorbent

The effect of CO_2_ concentration on the desulfurization penetration time was studied by varying the CO_2_ content in the gas mixture to 0%, 2.5%, 5.5% and 7.5%, where the reaction temperature was set at 80 °C and the inlet SO_2_ concentration was kept at 1200 mg/Nm^3^. The experimental results are shown in Figure 7a. With the increase in CO_2_ content, the penetration time of the adsorbent decreased significantly. Under the condition without CO_2_, the penetration time of the adsorbent reached 2500 s. With the addition of 2.5% CO_2_, it plummeted to 600 s. Further increasing the CO_2_ concentration, the penetration time did not drop proportional to the CO_2_ content.

Figure 7b shows the penetration sulfur capacity of desulfurization agents with different CO_2_ concentrations, and it can be seen that as the CO_2_ concentration increases, the overall sulfur capacity shows a decreasing tendency. When the CO_2_ content is 0%, the penetration sulfur capacity reached the maximum value of 6.5 g/100 g adsorbent, which means that the desulfurization effect of the adsorbent is the best at this time. When the CO_2_ content was increased to 2.5%, it was obvious that the penetration sulfur capacity significantly decreased to only 3.3 g/100 g desulfurization agent, and the penetration time was also shortened, indicating that the CO_2_ in the gas hinders the desulfurization process of the desulfurization agent and affects the desulfurization performance of it. When the CO_2_ content was further increased to 7.5%, there was a small decrease in the penetrating sulfur capacity, but not as drastic as before, indicating that CO_2_ has a decreasing effect on the absorption effect of desulfurization agent, and with the increase in CO_2_ concentration, there is a plateau, and the decreasing effect is weakened.

With the increase in CO_2_ concentration, the penetration time of the adsorbent was slightly shortened, the relative content of Ca(OH)_2_ decreased, and the relative content of CaCO_3_ increased, as shown in Figure 7c. The S content in the desulfurization ash was decreased, indicating that the desulfurization capacity of the adsorbent decreased.

Figure 8a shows the effect of different relative humidities on the desulfurization process. When relative humidity was increased from 0 to 20%, the penetration time was increased significantly. When the relative humidity was further increased, the penetration time was decreased. Correspondingly, the sulfur capacity at 20% relative humidity was the highest (Figure 8b). It can be concluded that an appropriate humidity is helpful for increasing the desulfurization effect of sorbents, as the desulfurization effect will be weakened by excessive water. The 20% relative humidity is appropriate in this study: the sulfur capacity of the adsorbent reaches 29 g/100 g.

### 3.4. Effect of Relative Humidity on Desulfurization Performance of Adsorbent

#### 3.4.1. Desulfurization Performance of Adsorbent

Under the condition of 20% relative humidity, the penetration time decreased with the increase in CO_2_ concentration (Figure 9a). The longest penetration time was 1260 s with 2.5% CO_2_ concentration, and the shortest penetration time was 580 s with 7.5% CO_2_ concentration. The overall sulfur penetration capacity tends to decrease as the CO_2_ concentration increases (Figure 9b). This is because the increase in CO_2_ concentration in the gas mixture consumes the adsorbent, resulting in a decrease in the adsorbent for desulfurization. This indicated that the CO_2_ concentration weakened the desulfurization effect of the desulfurizing agent at 20% relative humidity, and the sulfur penetration capacity gradually decreased as the CO_2_ concentration continued to increase.

#### 3.4.2. Morphology and Pore Structure of Desulfurized Ash

As shown in Figure 10a, the surface of the original adsorbent was rough and porous, resulting in a relatively large specific surface area (as shown in Table 3) that was favorable for gas adsorption. After desulfurization in 2.5% CO_2_ atmosphere (Figure 10b), the pores of the adsorbent became shallow, and there were many crystals growing on the surface. When the CO_2_ concentration increased to 5.5% (Figure 10c), the surface of the adsorbent obviously became dense and rough. The pores were filled, and the surface was covered with products which had thin strips, flakes, columns, needles and other irregular crystal structures. When the CO_2_ concentration was up to 7.5% (Figure 10d), the product crystal grew obviously on the adsorbent surface. As a result, the active points of the adsorbent were occupied and the pores were filled by the product, resulting in a decrease in BET and pore volume and pore diameter.

#### 3.4.3. Composition Analysis of Desulfurization Ash

The composition analysis of desulfurization ash is shown in Figure 11a. Without adding CO_2_, Ca(OH)_2_ accounted for about 76 wt.% of the desulfurization ash, indicating that the utilization rate of the adsorbent was not high. With the increase in CO_2_ content in the flue gas, the content of CaCO_3_ was increased, and the contents of Ca(OH)_2_ and CaSO_4_ and CaSO_3_ were decreased. This is consistent with the results of the TG analysis, as shown in Figure 11b. The weightlessness stages I (330~450 °C) and II (545~750 °C) represented the decomposed Ca(OH)_2_ and CaCO_3_. As we can see, with the increase in CO_2_ content, Ca(OH)_2_ was decreased and CaCO_3_ was increased. This is because the concentration of CO_2_ is much higher than that of SO_2_. Under the non-equilibrium condition, the formation trend of CaCO_3_ becomes larger.

From the above analysis, it can be seen that a large amount of CaCO_3_ was generated with high CO_2_ concentration in flue gas. To investigate whether in situ generated CaCO_3_ has a desulfurization effect, a group of experiments was carried out. Firstly, the adsorbent was put in the flue gas containing 5.5% CO_2_ atmosphere. When the CO_2_ in the exhaust gas reached stability, it meant the reaction had reached equilibrium, and at this time the surface of the sample was covered with produced CaCO_3_. Secondly, SO_2_ was introduced into the flue gas. In order to study the effect of CO_2_ on desulfurization, two comparative experiments were designed: one for the CO_2_ + SO_2_ gas mixture to examine the effect of CO_2_ on SO_2_ removal, and the other for cutting off CO_2_ gas and only passing SO_2_. The experimental results are shown in Figure 12. The penetration time of in situ generated calcium carbonate as the desulfurization agent under the condition of cutting off CO_2_ is 405 s, which is higher than that of the CO_2_ + SO_2_ gas mixture of 342 s but lower than that of Ca(OH)_2_ as the adsorbent. It can be seen that the in situ generated CaCO_3_ has a certain desulfurization ability, but the desulfurization effect is lower than that of Ca(OH)_2_.

It can be presumed that in the sintering flue gas with high CO_2_ concentration and strong oxygen potential, CaCO_3_ and CaSO_4_ are more easily generated, and the reaction can be divided into three stages as shown in Figure 13. At the first stage, CO_2_ and SO_2_ react with Ca(OH)_2_ at the same time to generate CaCO_3_ and sulfate, and Ca(OH)_2_ is continuously consumed as shown Equations (9)–(11). CaCO_3_ is consumed by reacting with SO_2_ as in Equation (12), while the generation rate is higher than consumption rate, so the amount of CaCO_3_ is increasing.

At the second stage, Ca(OH)_2_ is consumed, the surface is wrapped by desulfurization production, and the desulfurization by Ca(OH)_2_ is weakened. On the contrary, CaCO_3_ concentration is increased, and CaCO_3_ is the main desulfurization component as in Equation (12). After the peak point, the amount of CaCO_3_ is decreased. At the third stage, the export SO_2_ concentration exceeds the set value, and the adsorbent fails.
*CO_2_* + *Ca(OH)_2_* = *CaCO_3_* + *H_2_O*(9)
*SO_2_* + *Ca(OH)_2_* = *CaSO_3_* + *H_2_O*(10)
*CaSO_3_* + *O_2_* = *CaSO_4_*(11)
*SO_2_* + *CaCO_3_* = *CaSO_3_* + *CO_2_*(12)

After the implementation of the ultra-low emission standard in the steel industry, the limited emitting concentration of SO_2_ of sintering flue gas was decreased from the original 100 mg/Nm3 to 35 mg/Nm3. As shown in Figure 13, when the emission gas standards are reduced from S2 to S1, the penetration time of the absorbent is decreased from t2 to t1, so more fresh adsorbents are needed, and the corresponding CaCO_3_ content in the desulfurization ash C1 is also higher than that of C2, which is produced by the original emission standar

## 4. Conclusions

A series of experiments were carried out to study the effect of CO_2_ on the desulfurization of sintering flue gas with hydrated lime. The main conclusions can be summarized as follows: thermodynamic calculations showed that in the competitive relationship with Ca(OH)_2_, SO_2_ reacts more preferentially than CO_2_ under a thermodynamic standard state, and the desulfurization reaction is easier in the presence of oxygen. The penetration time of the adsorbent decreases with the increase in CO_2_ concentration, and the decrease is more obvious from the penetration of sulfur capacity, which indicates that CO_2_ is more likely to generate CaCO_3_ under the condition of moisture, thus impeding the desulfurization process. In situ generated calcium carbonate may have a desulfurization effect, but the desulfurization effect is not as good as Ca(OH)_2_.

However, a large proportion of CaCO_3_ was produced in the desulfurization ash under the condition that the CO_2_ concentration was absolutely dominant to SO_2_ in the sintering flue gas atmosphere.

## Figures and Tables

**Figure 1 materials-16-00303-f001:**
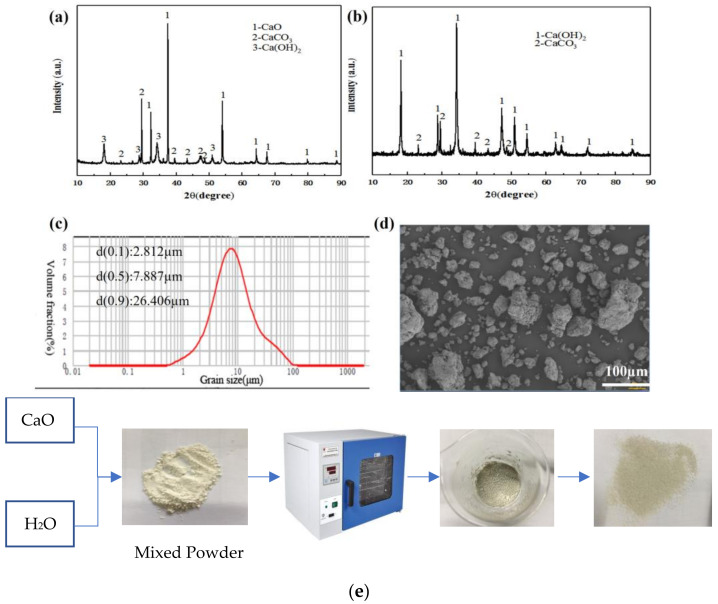
(**a**) XRD of raw quicklime; (**b**) XRD of digested sample; (**c**) particle size distribution of quicklime; (**d**) SEM; (**e**) the flow chart of the sample production processes.

**Figure 2 materials-16-00303-f002:**
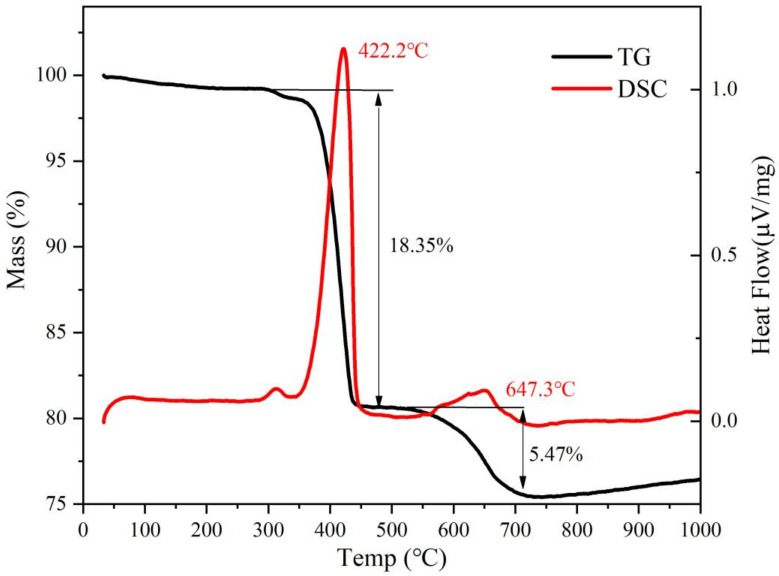
TG-DSC curves of raw quicklime.

**Figure 3 materials-16-00303-f003:**
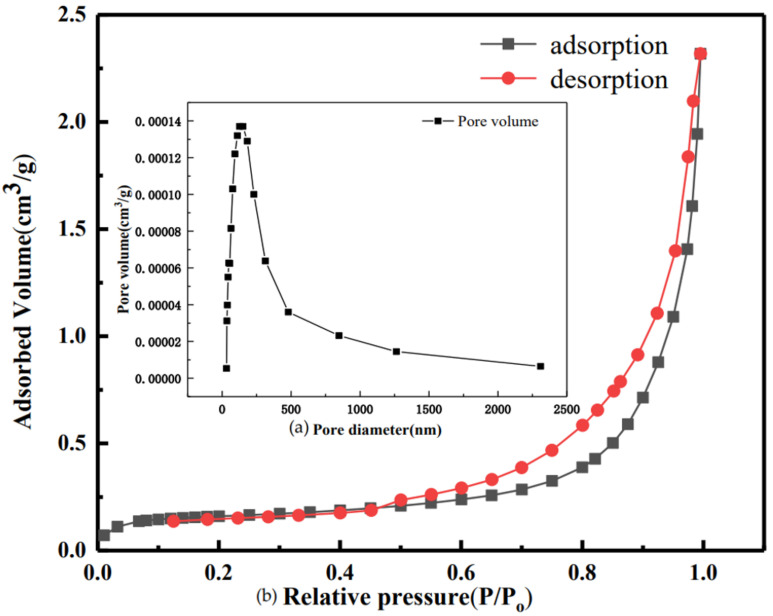
Isothermal adsorption and desorption curves (**a**) and pore size distribution curve (**b**) of quicklime.

**Figure 4 materials-16-00303-f004:**
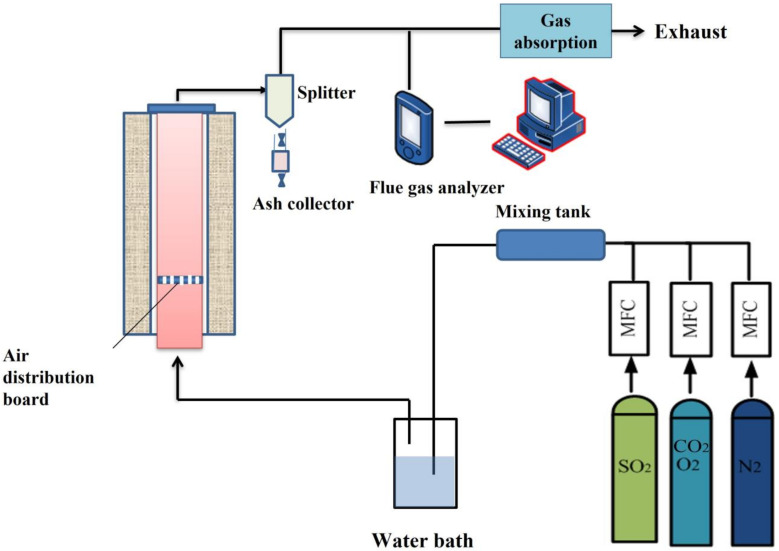
Schematic diagram of experimental system.

**Figure 5 materials-16-00303-f005:**
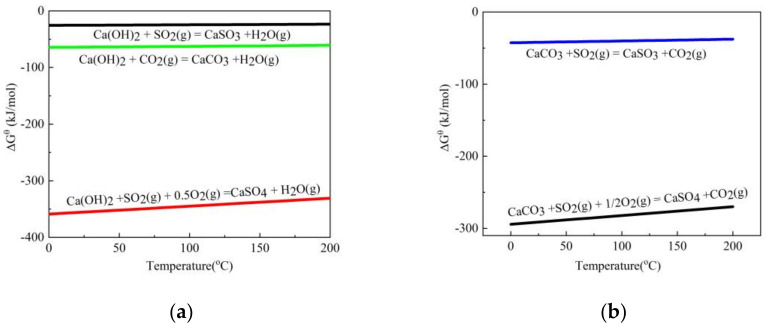
The ΔG^θ^-T diagram of chemical reactions of (**a**) Ca(OH)_2_ and mixed gas; (**b**) CaCO_3_ and mixed gas.

**Figure 6 materials-16-00303-f006:**
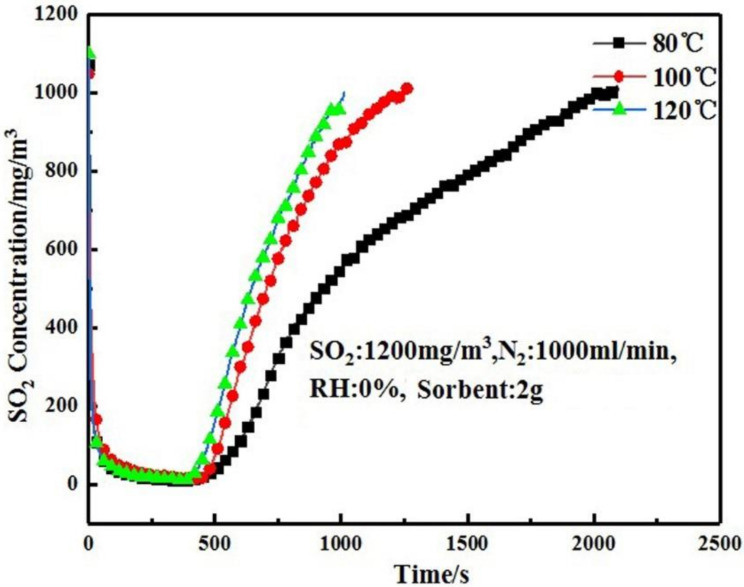
Effect of temperature on desulfurization performance of adsorbent (RH = 0%, SO_2_ = 1200 mg/Nm^3^).

**Figure 7 materials-16-00303-f007:**
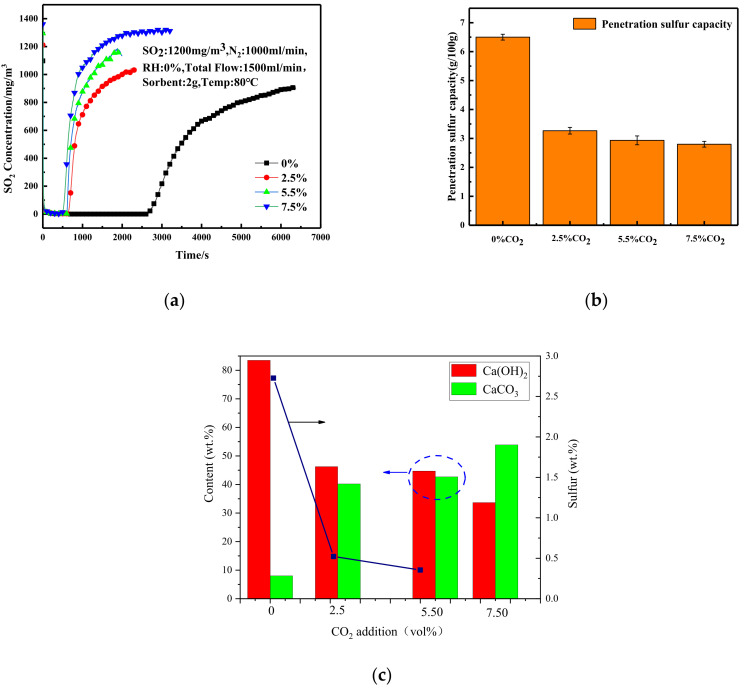
Effect of CO_2_ concentrations on (**a**) desulfurization performance; (**b**) sulfur penetration capacity; (**c**) relative content of Ca(OH)_2_ and CaCO_3_ in desulfurization as4. Effect of relative humidity on desulfurization performance of adsorbent

**Figure 8 materials-16-00303-f008:**
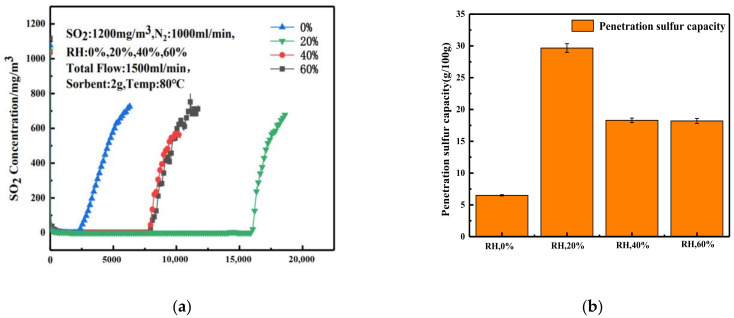
Effect of relative humidity on (**a**) the desulfurization performance and (**b**) penetration sulfur capacity of desulfurization agent.

**Figure 9 materials-16-00303-f009:**
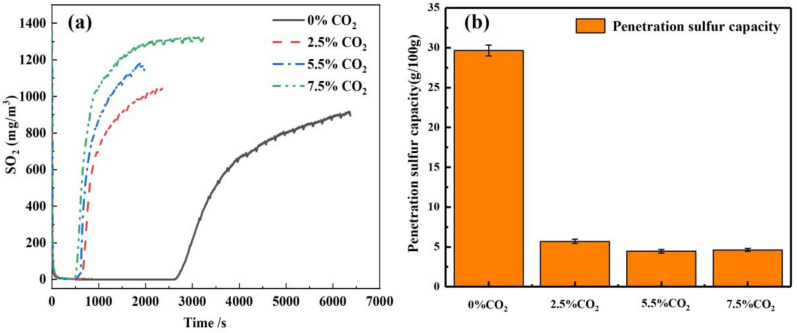
(**a**) Effect of different CO_2_ contents on desulfurization time; (**b**) sulfur penetration capacity of adsorbents at 20% relative humidity.

**Figure 10 materials-16-00303-f010:**
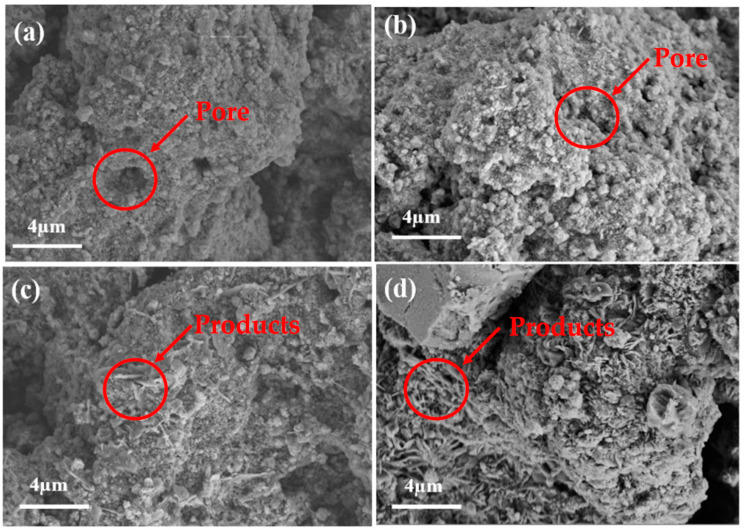
Surface morphology of adsorbent: (**a**) original desulfurization; (**b**) 2.5%; (**c**) 5.5%; (**d**) 7.5%.

**Figure 11 materials-16-00303-f011:**
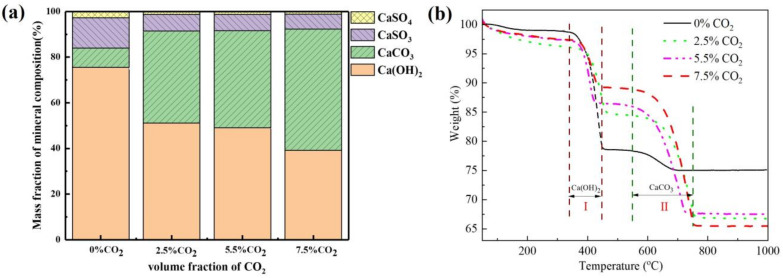
(**a**) Composition analysis; (**b**) TG curves of desulfurized ash under different CO_2_ atmosphere6. The mechanism of CaCO_3_ on the desulfurization process

**Figure 12 materials-16-00303-f012:**
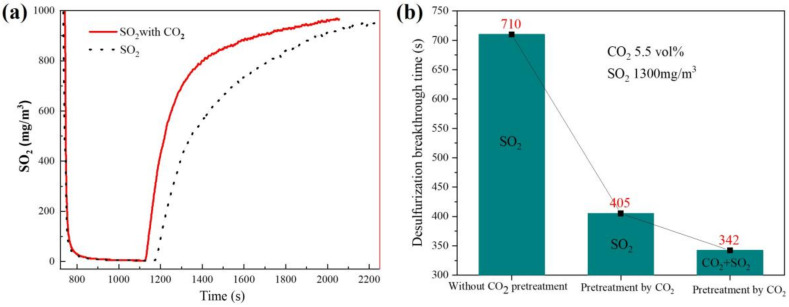
Effect of CaCO_3_ on (**a**) desulfurization and (**b**) desulfurization penetration time under different conditions.

**Figure 13 materials-16-00303-f013:**
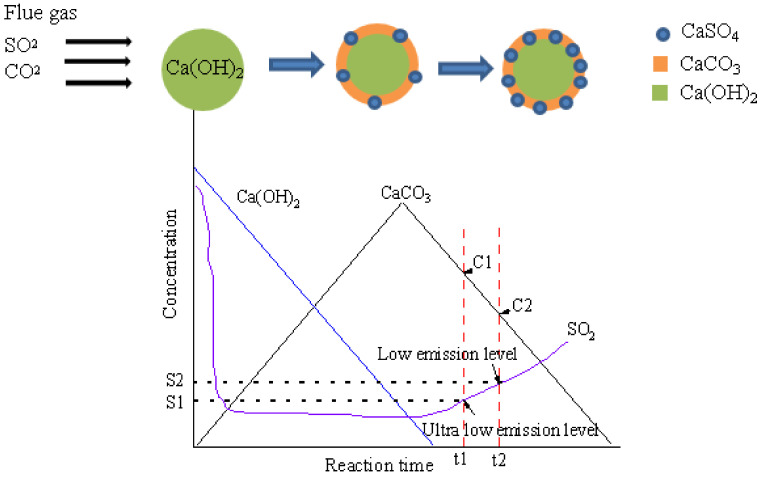
Schematic diagram of desulfurization process of sintering flue gas.

**Table 1 materials-16-00303-t001:** XRF element analysis of dried quicklime samples (wt.%).

Elements	Ca	Mg	Fe	Al	Si	K
Wt.%	88.26	0.83	0.62	0.47	0.35	0.23

**Table 2 materials-16-00303-t002:** Gas composition and flow rate

Gas Type	Mixed Gas	SO_2_	N_2_
Concentration	10%CO_2_ + 5%O_2_ + surplus N_2_	4200 ppm	99.99%

**Table 3 materials-16-00303-t003:** BET and pore structure parameters of desulfurization products with different CO_2_ concentrations.

Samples	BET Specific Surface Area/m^2^∙g^−1^	Average Pore Size/nm	Pore Volume/cm^3^∙g^−1^
Original adsorbent	11.74	166.14	0.049
0% CO_2_	10.87	126.18	0.036
2.5% CO_2_	7.56	163.12	0.033
5.5% CO_2_	8.91	137.21	0.033
7.5% CO_2_	7.63	136.30	0.028

## Data Availability

The data presented in this study are available on request from the corresponding authors. The data are not publicly available due to privacy or ethical restrictions.

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
