# Peer review of "Effect of CO2 on the Desulfurization of Sintering Flue Gas with Hydrated Lime"

_materials, 2022, doi:10.3390/ma16010303_

Round 1
Reviewer 1 Report
An interesting manuscript on reducing pollution from flue gases. The authors should consider the comments below for revision.
-The Introduction needs editing for English. The remainder of the manuscript was fairly well-written.
-Line 31-33. It would appear that the emission standard was decreased rather than increased, so I don't think "raised" is the right word here.
-Line 70-72. The authors should include in this paragraph a discussion of how this article advances the state-of-the-art. What is new or novel about this study relative to the existing literature?
-Line 80. If the quicklime sample contains CaCO3, then some it is has been carbonated and will not be effective in this study. The authors should provide quantitative analysis to demonstrate the purity of the sample. Quantitative XRD, thermogravimetric analysis, or complexometric titration after ethylene glycol digestion are all methods could be used to determine this.
-Line 97. What was the purity of the gases?
-Line 163. What purge gas was used for TG-DSC experiments and what heating rate?
-Equation 4. You state on Line 111 that Ca(OH)2 does not react with SO2 as a solid, but Equation 4 shows Ca(OH)2 as a solid. If the reaction proceeds through an aqueous solution, should this equation be rewritten to reflect that?
-Figure 6. Please increase font size or figure size. The figures are very hard to read.
-Figure 7. Please increase font size or figure size. The figures are very hard to read.
-Figure 8. Please increase font size or figure size. The figures are very hard to read.
-Figure 9. Perhaps include arrows point to explicitly what is discussed regarding this figure. These SEM images are difficult to interpret; the morphology in each of the 4 images appears identical.
-Figure 10. Please increase font size or figure size. The figures are very hard to read.
-Figure 11. Please increase font size or figure size. The figures are very hard to read.
-Equation 12. The authors are proposing that SO2 reacts with CaCO3. Where is the evidence of this? CaCO3, if it precipitates, has a fairly low solubility, so what is the probability of Ca2+ in solution to react with SO2? Furthermore, why could CO2 be released? If CaCO3 is dissolved in solution, then then CO2 exists as the anion CO3 2-. This proposed mechanisms appears unconvincing, unless the authors can provide more data or provide support from the literature.
Reviewer 2 Report
Dear Editor,
I'm very glad to be the review of this paper (Manuscript Number ID: materials-2039435). Title: Effect of CO2 on the desulfurization of Sintering Flue Gas with hydrated lime. I decided to accept this paper for the publishing in Journal of Materials (ISSN 1996-1944) after major correction with following comments:
1. I could not found a recognizing contribution in this paper there are plenty of results, but the authors were not focusing on the novelty of the paper.
2. I have not found a comparative study with other absorbent such as ionic liquid.
Chemical Engineering Research and Design 182 (2022) 659–666. https://doi.org/10.1016/j.cherd.2022.03.047.
3. English should be improved throughout the manuscript.
4. The abstract must be rewritten again with a reduction and explaining the major finding with a conclusion.
5. The introduction part must be containing modern references such as:
Microporous and Mesoporous Materials 341 (2022) 112020. https://doi.org/10.1016/j.micromeso.2022.112020.
6. The aims of the present work must be more cleared at the end of the introduction part. Chemistry Africa. https://doi.org/10.1007/s42250-022-00447-9.
7. The solubility hydrate (Ca(OH)2) as absorbent is very important. The author must be discussing this point in detail because the solubility is a very important factor in increasing the reaction rate.
8. I hope the author draws the schematic diagram for all the processes.
9. What is the relationship between surface tension and solubility for the hydrate (Ca(OH)2) as absorbent. The author must be discussing this point in detail.
10. The author must be clear about the experimental steps and discuss the interfacial phenomena for this study.
11. What is the main conclusion of this study? The conclusion must be reduced.
12. Maybe the leaching occurred from the surface of the hydrate (Ca(OH)2) as absorbent materials that mean dissolved in the solution. I need the author to explain this important point in the result and discussion part.
13. The hydrate (Ca(OH)2) as absorbent material characterizations must appear.
14. Why the author used a hydrate (Ca(OH)2) as absorbent.
15. I hope the author of this manuscript achieves a comparison between this study and others the same reaction.
16. What is the purpose from Kinetics study such as; pseudo-first, second-order, and Intraparticle diffusion was included in this study.
17. The author must be clear the experimental steps for the preparation of hydrate (Ca(OH)2) as absorbent and stock solution.

Round 2
Reviewer 1 Report
The authors have provided a revised manuscript and have addressed this reviewer's comments. However, some additional comments and suggestions remain:
-Line 71-78. The authors should include a clear statement about what is new or novel about this study relative to the literature. This is still unclear.
-Line 95. I cannot follow the logic how the authors derived the CaCO3 content of 4.76%. There is no suggestion from Table 1 how this is determined. TGA would be able to easily determine the Ca(OH)2 and CaCO3 contents in the lime sample. The authors should make an effort to compute the Ca(OH)2 and CaCO3 contents in the lime sample to establish the purity of the lime.
-FIgure 4. Please include the font sizes. This figure is difficult to read since the text is so small.
-The authors did not modify the article in response to Point #14 in the previous version. This was the comment regarding the validity of Equation 12. The authors provide an argument that Equation 12 occurs, which is still unclear based on the data. The authors comment that this will be addressed in a future study, which is an acceptable answer, but then the authors should modify the discussion in the present study to limit the speculation. If the data show that SO2 is removed, where is the evidence that CaCO3 converts to CaSO4? Perhaps the SO2 simply sorbs to the CaCO3 particle surface? The authors should really limit the speculative discussion without evidence.
Reviewer 2 Report
Accept in present form for publication
Author Response
Thank you for your acceptance!